# communications
## engineering

# Polarization-independent wavelength demultiplexer based on a single etched diffraction grating device

Chenguang Li [1], Bo Xiong[1] & Tao Chu [1✉]

Polarization-compatible receivers are indispensable in transceivers used for wavelength division multiplexing optical communications, as light polarization is unpredictable after transmission through network fibers. However, the strong waveguide birefringence makes it difficult to realize a polarization-independent wavelength demultiplexer in a silicon photonic receiver. Here, we utilized the birefringence effect for simultaneously demultiplexing wavelengths and polarizations, and experimentally demonstrated a polarization-independent wavelength demultiplexer with a single device on a SiPh platform. The principle was validated on an etched diffraction grating, which successfully split the arbitrarily polarized light containing four wavelengths into eight channels with single-polarization and single-wavelength signals. Polarization-dependent losses of 0.5–1.8 dB, minimum insertion loss of 0.5 dB, and crosstalks lower than −30 dB were experimentally measured for the fabricated etched diffraction grating. Thus, a promising general solution was developed for implementing polarization-independent wavelength division multiplexing receivers and other polarization-independent devices on photonics platforms with birefringent waveguide devices.

---

[1] College of Information Science and Electronic Engineering, Zhejiang University, Hangzhou, China. ✉email: chutao@zju.edu.cn

Wavelength division multiplexing (WDM) is one of the most important technologies for optical communications with its outstanding advantages in terms of low cost and high capacity[1,2], which overcomes the transmission limit of a single optical fiber by carrying signals with multi-wavelength light[3,4]. Wavelength multiplexers and demultiplexers combine and separate multi-wavelength lights, respectively, and are the key devices of WDM systems. However, owing to the change in the state of polarization (SOP) of light in the network fibers, the demultiplexers must be able to operate in arbitrary SOPs to avoid the serious problem of polarization-dependent loss (PDL)[5,6]. Usually, polarization-independent demultiplexers can be easily fabricated on a $SiO_2$-based planar light circuit (PLC) platform; unfortunately, such devices have large footprints and cannot be integrated with photodiodes (PDs) monolithically, leading to a reduction in integration density[7–9]. In contrast, InP-based demultiplexers can be monolithically integrated with PDs and are polarization independent[10,11]. However, they still suffer from large footprints and high costs. Lately, silicon photonic (SiPh) integration is being considered a very promising platform for constructing WDM systems with the benefits of high-density monolithic integration, low cost, and compatibility with complementary metal-oxide-semiconductor processing[12]. However, owing to the structural birefringence in silicon-based waveguides, the operation of demultiplexers on a SiPh platform is usually restricted to a single polarization[13–19].

Polarization diversity schemes[20–24] are effective for solving the device polarization dependence issue on a silicon-based platform, which needs two similar sets of light circuits for signal processing. In this system, light with arbitrary SOPs is divided into TE (transverse electric) and TM (transverse magnetic) polarizations; then, they are converted to the same TE/TM polarizations and transmitted separately to two similar sets of light circuits for processing. Finally, they are converted back to different TE/TM polarizations for combination and output. Recently, researchers have developed more simplified polarization-independent demultiplexers by cascading a single demultiplexer with polarization-handling devices such as a polarization beam splitter[25,26] or a series of polarization rotators[27,28]. In these devices, the polarization dispersions are compensated with the angular dispersion via polarization beam splitters or polarization rotation via polarization rotators. To reduce the insertion loss, other polarization-independent demultiplexer designs have been proposed. One is based on a thick-top-silicon-based platform[29], in which low polarization dependence can be achieved with large cross-sectional waveguides. Further, a nanostructured free propagation region (FPR) has been used for polarization dispersion compensation[30,31]. However, there are no reports on achieving polarization independence balancing a small footprint, simple process, and low loss. Thus, handling the received light beams with arbitrary SOPs for wavelength demultiplexing remains an unsolved serious issue for SiPh devices; it requires a separate large silica PLC demultiplexer chip, which has to be packaged in a commercial transceiver along with monolithic integrated SiPh chips[8,9].

Here, we propose a polarization-independent wavelength demultiplexer based on a single SiPh etched diffraction grating (EDG) device. By utilizing the difference in the effective refractive index between the TE and TM polarizations in high-index-contrasted slab waveguides, lights with various polarizations and wavelengths can be transmitted to different output channels on the EDG Roland circle. Thus, the demultiplexing of the wavelengths and polarizations can be realized in a single EDG device simultaneously. Based on this idea, a polarization-independent EDG demultiplexer for a coarse WDM (CWDM) was fabricated and verified on a silicon nitride ($Si_3N_4$) thin-film platform. The measurement results showed that the EDG had insertion losses ranging from 0.5 dB to 2.4 dB, crosstalks below −30 dB, and PDLs between 0.5 and 1.8 dB at four target wavelengths.

Moreover, a scan of the SOP of the incident light revealed that the outputs at four wavelengths remained almost stable with small standard deviations of 0.4–0.5 dB for the equator and 0.2–0.3 dB for the longitude of the Poincaré sphere. These results prove that a polarization-independent wavelength demultiplexer with low loss was successfully realized with a single EDG device. This work provides a general solution for polarization-dependent WDM transceivers in SiPh and other platforms with birefringent waveguide devices.

## Results

**Principle and design**. The schematic of the proposed polarization-independent EDG demultiplexer is shown in Fig. 1a. An EDG performs wavelength multiplexing and demultiplexing by using the phase difference induced by the wavelength-dependent refractive index. Owing to the internal birefringence, the effective refractive indices of the TE and TM polarizations usually have differences, which result in EDG polarization dependence. However, in our design, this polarization dispersion was utilized as a new degree of freedom to build a multi-dimensional demultiplexer useful for both polarizations and wavelengths. By elaborately designing the polarization dispersion to achieve the phase matching of the EDG in the polarization dimension, light beams with different polarizations can be separated in the Roland circle to realize polarization demultiplexing based on wavelength demultiplexing. The proposed EDG device is fabricated on a $Si_3N_4$ thin-film platform consisting of four main components, namely, input waveguide for input signal light, FPR for light transmission, etched gratings for reflecting and converging multi-wavelength multi-polarization signal lights, and TE/TM output waveguides for output ideal signal lights. In addition, one-dimensional Bragg gratings[32,33], which are typically used to increase the reflection efficiency of EDG teeth, are added behind the grating teeth. When light with multi-wavelengths in arbitrary SOPs is input into the specially designed EDG, it will be demultiplexed into multiple light beams with single polarization (TE/TM) and single wavelength, and the outputs will be generated in different channels depending on the polarization and wavelength. By feeding the output TE and TM light beams of the same wavelength into one PD, wavelength demultiplexing is achieved with polarization independence in a single EDG device.

The design of an EDG is commonly based on the principles of blazed grating and Roland mounting[34,35]. Our EDG is first identified with TE polarization and its operation is then analyzed in TM polarization, as shown in Fig. 1b. The grating teeth located at the tangential points of the Rowland circle (green dotted line) and grating circle are considered as the central grating teeth ($0^{th}$). $L_f$ is the Rowland circle diameter, $L_{0in}$ and $L_{0out}$ are the effective incident and output optical paths of the central grating tooth, respectively, $L_{kin}$ and $L_{kout}$ are the effective incident and output optical paths of the $k_{th}$ grating tooth, respectively, and $\theta_{in}$ and $\theta_{out}$ are the incidence and diffraction angles, respectively. $\theta_k$ is the rounding angle of the grating circle corresponding to the midpoint of each grating tooth. The EDG needs to satisfy the optical path difference, as shown in Eq. (1), and thus, $\theta_{out}$ satisfies the relationship with the effective refractive index $n_{eff}$ of the FPR slab waveguide, as shown in Eq. (2), which is polarization-dependent.

$$(L_{kin} + L_{kout}) - (L_{0in} + L_{0out}) = \frac{km\lambda}{n_{eff}}, \qquad (1)$$

$$(L_{kin} + L_f \cdot \sqrt{(\sin\theta_k + \cos(\theta_{out}) \cdot \sin(\theta_{out}))^2 + (\cos\theta_k + \sin(\theta_{out})^2)^2}) - (L_{0in} + L_f \cdot \cos(\theta_{out})) = \frac{km\lambda}{n_{eff}}, \qquad (2)$$

where $m$ is the diffraction order, $\lambda$ is the incident light wavelength, and $n_{eff}$ is the effective index of the propagation medium, which is

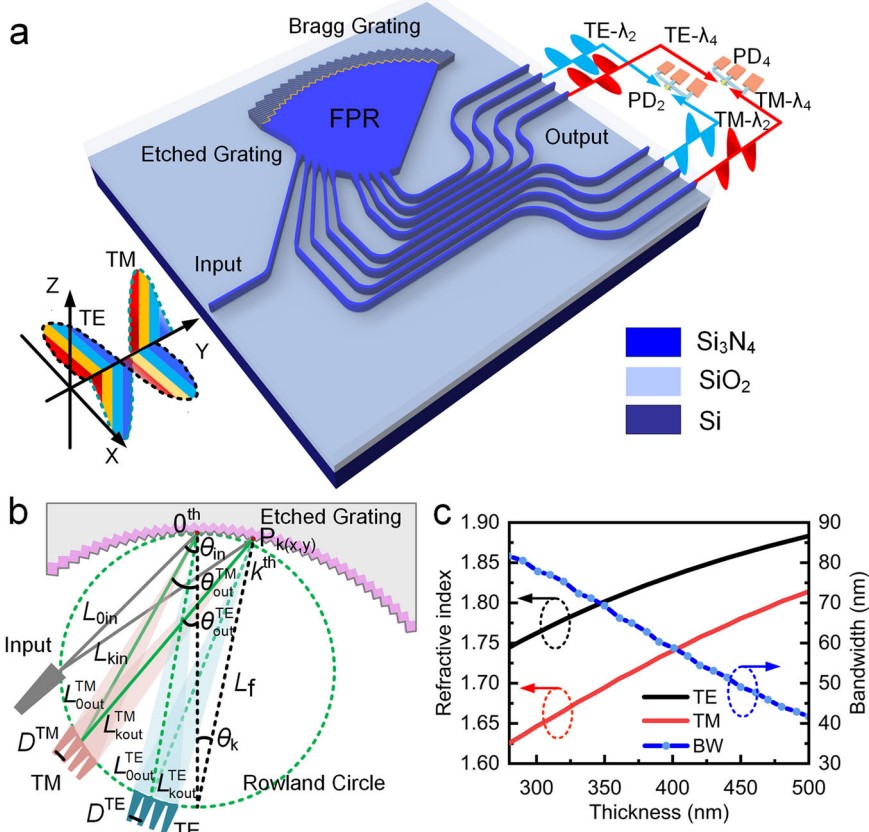

**Fig. 1 Device architecture and design principle. a** Schematic of polarization-independent etched diffraction grating (EDG) demultiplexer. When multiplexed multi-wavelength light beams in arbitrary states of polarization are input, they are demultiplexed into light beams with single transverse electric (TE)/transverse magnetic (TM) polarization and single wavelength in free propagation region (FPR) and output on different channels. **b** Schematic of polarization and wavelength separation. **c** Effective refractive indices of TE/TM polarizations and working bandwidth (BW) of polarization separation corresponding to the thicknesses of the $Si_3N_4$ layer at 1310 nm wavelength.

usually considered as the effective refractive index of the fundamental mode of the slab waveguide. In Eq. (2), $L_{kin}$, $L_{0in}$, $\theta_k$, $k$, $m$, $\lambda$, and $L_f$ are definite and independent of the polarization (see Supplementary Note 1 and Figs. S1 and S2). Thus, the light output angle $\theta_{out}$ can be determined by the effective refractive index $n_{eff}$ of the FPR slab waveguide. Owing to the large difference in $n_{eff}$ between the TE and TM polarizations, $\theta_{out}$ of the TE- and TM-polarized light beams are different, which indicates that TE and TM light beams as well as lights of different wavelengths are output at different positions of the Roland circle, that is, light signals of different polarizations and wavelengths can be completely separated.

Further design is needed for separating the polarizations in the desired wavelength range. The polarization dispersion in the FPR slab waveguide can be regulated by changing its material and thickness. In this study, $Si_3N_4$ was selected because of its low propagation loss, low thermo-optical effect, and complementary metal-oxide-semiconductor processing compatibility[28,36]. The effective refractive indices of the TE and TM fundamental modes in the FPR slab waveguides of different $Si_3N_4$ layer thicknesses were simulated, as shown in Fig. 1c. As the $Si_3N_4$ layer becomes thinner, the refractive index difference between the TE and TM polarizations increases, resulting in a larger separation between output angles under the TE and TM polarizations according to Eq. (1). In this case, the working bandwidth in the O-band, which is defined as the range of wavelength when the outputs of the two polarizations do not overlap, also increases a lot, as shown in the calculated results in Fig. 1c. However, to achieve lower loss and crosstalk of the EDG demultiplexer, Bragg gratings requires a

thicker $Si_3N_4$ layer to maintain a sufficient reflection bandwidth. We simulated the Bragg gratings reflection bandwidth with different $Si_3N_4$ layer thicknesses and verified that a thickness of more than 280 nm can maintain a sufficient reflection bandwidth in the wavelength range of 1270–1340 nm, while polarization separation dispersion requires a $Si_3N_4$ layer thickness of less than 325 nm. Thus, a balanced $Si_3N_4$ layer thickness was obtained as 310 nm (details can be found in Supplementary Note 2 and Fig. S3). Moreover, the input and output waveguides were connected to the FPR with the tapered waveguides to reduce coupling loss. Several geometric parameters were carefully optimized, including the Bragg grating period (418 nm), diffraction order ($m = 3$), and output waveguide spacing ($d = 5\,\mu m$) (details can be found in Figs. S3–S6 in Supplementary Note 2). Finally, a low-loss O-band CWDM demultiplexer was obtained for polarization-independent wavelength demultiplexing.

**Simulation results**. The operation of the polarization-independent EDG demultiplexer was verified through simulations with the Lumerical 2.5D finite-difference time-domain software, as shown in Fig. 2. When the multi-wavelength multiplexed light with TE polarization was input, the demultiplexed light beams were output from channels 1–4, whereas channels 5–8 generated very less output. On the contrary, in the case of TM polarization, light beams were output from channels 5–8 whereas limited output was obtained from channels 1–4. This fact shows that the TE/TM polarizations were demultiplexed along with the wavelengths in the EDG. The insertion losses of the channels were

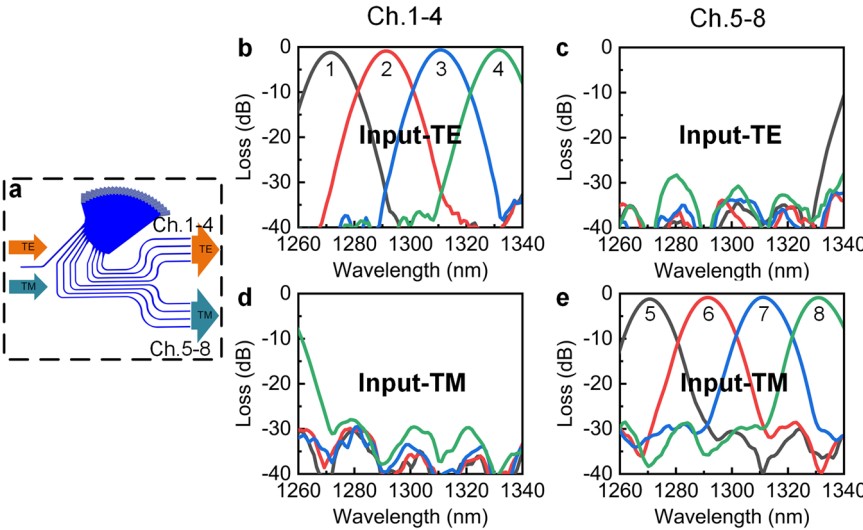

**Fig. 2 Simulated transmission spectrum of polarization-independent demultiplexing EDG. a** Schematic of EDG model. **b** Output from channels (Ch.) 1–4 with TE input. **c** Output from Ch. 5–8 with TE input. **d** Output from Ch. 1–4 with TM input. **e** Output from Ch. 5–8 with TM input.

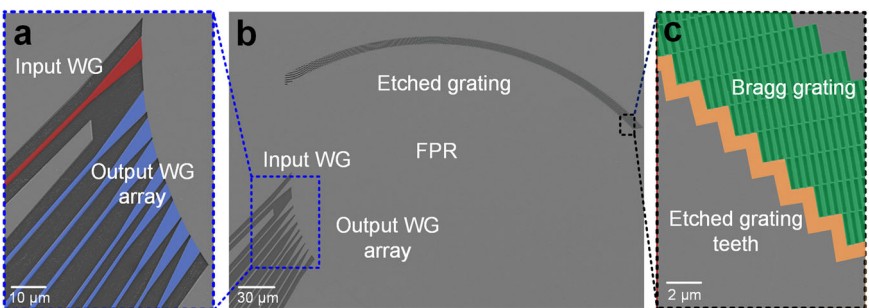

**Fig. 3 Scanning electron microscope (SEM) image of EDG after ICP etching. a** SEM images of waveguide (WG) array attached to the FPR. **b** SEM of the overall structure of polarization-independent demultiplexing EDG. **c** SEM of boundary of etched grating teeth.

estimated to be 1.2 dB (1271.1 nm), 0.9 dB (1291.7 nm), 0.6 dB (1311.1 nm), and 0.7 dB (1331.0 nm) for the TE-polarized signals, whereas the values for the TM-polarized signals were 1.2 dB (1271.1 nm), 0.9 dB (1291.7 nm), 0.8 dB (1311.1 nm), and 0.9 dB (1331.0 nm). The device demonstrated a good loss uniformity (approximately 0.5 dB) and low crosstalk (better than −28 dB) which is mainly caused by the coupling of adjacent channels. The PDL was less than 0.2 dB, which indicates excellent polarization-independent performance. It should be noted that owing to the TE and TM polarizations being separated into different channels, the output positions for the TE and TM polarizations can be adjusted independently according to the linear dispersion on the Roland circle. Thus, a larger degree of design freedom can be obtained, and the dependence of the EDG wavelength shift on the polarization can be easily reduced to nearly zero.

**Measurement results of the fabricated device**. The proposed polarization-independent demultiplexers were fabricated and tested. The details of the fabrication and measurement processes are described in the Methods section. The footprint of the fabricated EDG is $320 \times 230\ \mu m^2$, and the scanning electron microscope (SEM) images of the device are shown in Fig. 3. Figure 3a, c are magnified views of the rectangular areas in Fig. 3b. Figure 3a depicts the input and output waveguides connected to the FPR with tapered waveguides, while Fig. 3c depicts the Bragg gratings behind the etched grating teeth.

The EDG demultiplexer was characterized using the experimental setup shown in Fig. 4a. The normalized transmission spectra in Fig. 4c–f shows that the EDG successfully separated the TE and TM multiplexed multi-wavelength lights into different output channels, which agree with the simulation results shown in Fig. 2. The peak output light wavelengths were 1284.2 nm, 1303.9 nm, 1322.9 nm, and 1342.3 nm for the TE-polarized input and 1283.2 nm, 1303.5 nm, 1322 nm, and 1341 nm for the TM-polarized input. Compared with the simulations, around 12 nm red shifts of the output wavelengths were observed, which were considered to arise from the unexpected change in thickness of the $Si_3N_4$ layer during the $Si_3N_4$ film deposition. According to the error analysis (shown in Supplementary Note 3 and Fig. S7), a change in thickness of approximately 10 nm caused a shift of 6–8 nm (for the 310-nm-thick $Si_3N_4$ layer) in the wavelength in the EDG output due to a change in the effective refractive index. And the measured thickness of the $Si_3N_4$ layer showed a variation of approximately 19.2 nm in this study, which agreed well with the simulation. When the $Si_3N_4$ layer thickness increases to 500 nm, the wavelength shift from a 10 nm thickness change is reduced to 3–4 nm. However, due to the bandwidth range required by the CWDM4 design (shown in Fig. S8), we could only choose a thinner $Si_3N_4$ layer. Nevertheless, the EDG wavelength intervals remained close to the designed value of 20 nm. The insertion losses of the channels were 2.4 dB, 2.3 dB, 2.3 dB, and 1.3 dB for the TE-polarized input, and 1.9 dB, 1.1 dB, 0.5 dB, and 2.2 dB for the TM-polarized input, with PDLs of 0.5–1.8 dB. The crosstalks were better than −30 dB for both TE and TM polarizations.

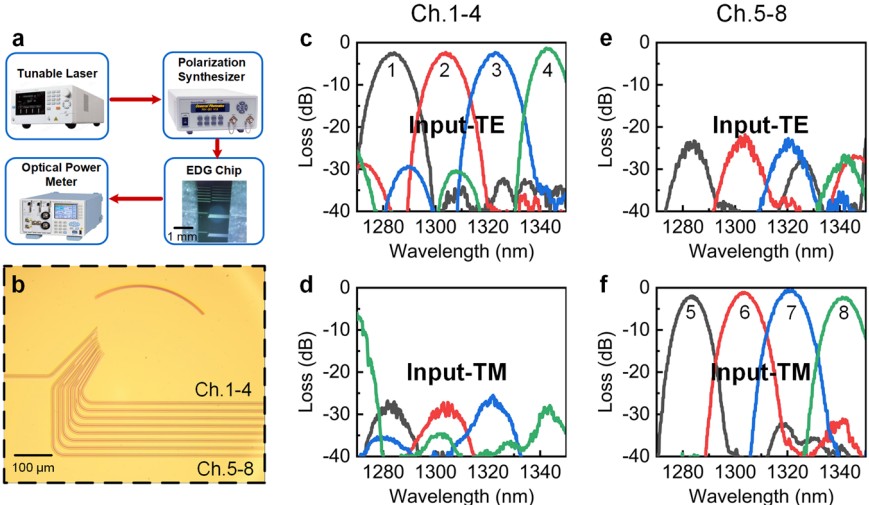

**Fig. 4 EDG measurements. a** Experimental setup. **b** Optical microscope view of the fabricated EDG. **c–f** Transmission spectra of wavelength scans at the eight channels under TE polarization and TM polarization. **c** Outputs from Ch. 1–4 with TE input. **d** Outputs from Ch. 1–4 with TM input. **e** Outputs from Ch. 5–8 with TE input. **f** Outputs from Ch. 5–8 with TM input.

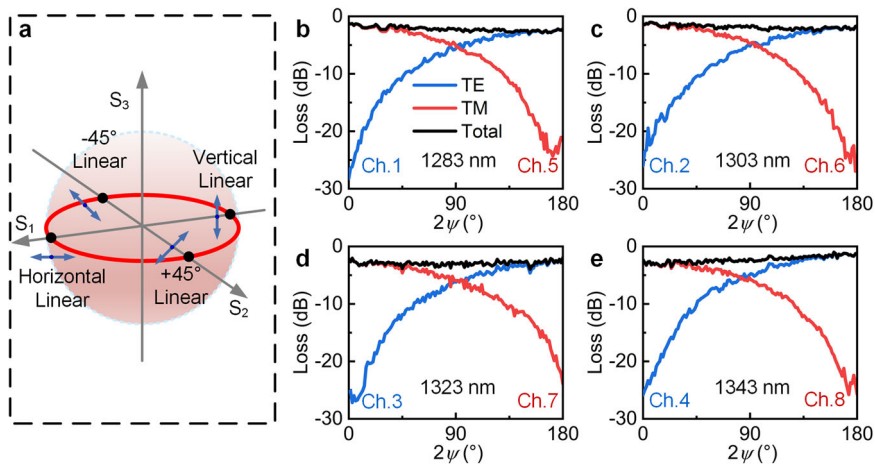

**Fig. 5 Scanning measurement along the RHC trajectory (red line) with azimuthal angles 2ψ∈(0, π) on the horizontal axis and losses on the vertical axis. The outputs (black solid) are superimposed by the TE (blue solid) and TM (red solid) outputs of the same wavelength. a** Polarization state represented by Poincaré sphere. **b–e** Transmission spectra of polarization scans along the RHC trajectory at the output wavelengths of the eight channels. **b** Ch. 1 and Ch. 5 at 1283 nm. **c** Ch. 2 and 6 at 1303 nm. **d** Ch. 3 and Ch. 7 at 1323 nm. **e** Ch. 4 and Ch. 8 at 1343 nm.

Compared with the simulations, the increased insertion losses and PDLs mainly resulted from the perpendicularity and roughness of the sidewalls of the Bragg gratings, which can be improved by further optimization of the fabrication process.

**Device testing in arbitrary SOPs.** To test the performance of the EDG device in arbitrary SOPs, the SOP of the input light was scanned with a polarization synthesizer, which could generate the desired SOP signals at fixed wavelengths. The receding horizontal control (RHC) trajectory (see Fig. 5a) was scanned so that the SOP of the input light evolved from TM linear polarization to TE linear polarization, with the azimuth angle changing gradually from 0° to 180°. The results are shown in Fig. 5b–e. Despite the TE and TM output intensities showing large changes with opposite trends according to Malus's law, the total outputs of the two polarization channels at the same wavelength (black solid) were close to constant values of 2.21 dB, 1.85 dB, 2.88 dB, and 2.27 dB with standard deviations of 0.39 dB, 0.35 dB, 0.37 dB, and 0.54 dB, respectively, for the four wavelengths. This indicates that

the optical signal that the PD received from the superimposed light remained stable at arbitrary linearly polarized input lights.

Moreover, the influence of the polarization phase on the device was investigated, as shown in Fig. 6. The polarization phase of the input light was changed so that the input SOP changed from −45° linear polarization through elliptical polarization and circular polarization and then to 45° linear polarization along the LP0 trajectory, whereas the intensities of the TE and TM polarizations were fixed and equal. Figure 6 shows that the superimposed outputs of the TE and TM polarizations remained stable at 1.75 dB, 1.58 dB, 2.13 dB, and 2.12 dB for the four wavelengths with variation in the polarization phase, with standard deviations of 0.29 dB, 0.31 dB, 0.20 dB, and 0.30 dB, respectively. This shows that the change in the polarization phase has a weak effect on the device's operation. Hence, the EDG can operate independently with any linearly, elliptically, and circularly polarized light input. Based on the experiments shown above, a polarization-independent wavelength demultiplexer with high efficiency was successfully demonstrated with a single EDG device.

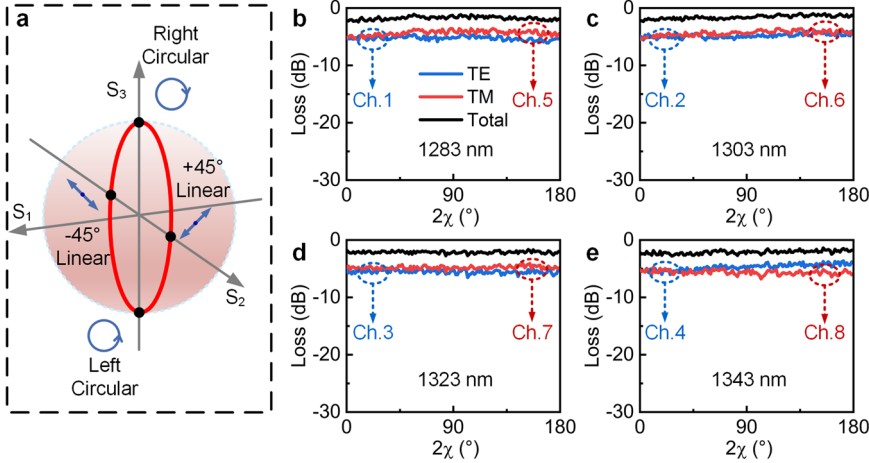

**Fig. 6 Scanning measurement along the LP0 trajectory (red line) with ellipticity (0, π) on the horizontal axis and device loss on the vertical axis.** **a** Polarization state represented by Poincaré sphere. **b–e** Transmission spectra of polarization scans along the LP0 trajectory at the output wavelengths of the eight channels. **b** Outputs of Ch. 1 and Ch. 5 at 1283 nm. **c** Outputs of Ch. 2 and Ch. 6 at 1303 nm. **d** Outputs of Ch. 3 and Ch. 7 at 1323 nm. **e** Outputs of Ch. 4 and Ch. 8 at 1343 nm.

## Discussion

This paper proposed a design of polarization-independent wavelength demultiplexer by utilizing the birefringence effect, which can be extended to numerous planar wavelength demultiplexing devices such as EDGs and arrayed waveguide gratings. The design principle was successfully applied on an EDG device on a silicon-based $Si_3N_4$ platform, and a polarization-independent wavelength EDG demultiplexer was experimentally fabricated and demonstrated. Measurements by scanning the polarizations of input lights showed that stable outputs could be obtained at four demultiplexed wavelengths with standard deviations of 0.4–0.5 dB for the equator of the Poincaré sphere and 0.2–0.3 dB for the longitude of the Poincaré sphere, which demonstrated the good polarization-independent property of the EDG device. The insertion losses of the EDG device were measured to be 0.5–2.4 dB with PDLs of 0.5–1.8 dB, and the crosstalk was better than −30 dB at the four output wavelengths. The comparison of the device proposed in this paper with other WDM devices can be found in the Supplementary Note 4 and Table S1. This compact and low-loss polarization-independent wavelength demultiplexer, which can be monolithically integrated with PDs and other devices on a SiPh platform, has broad applicability in telecommunication and data communication systems. The design scheme of using the birefringence effect, proposed by us, not only simplifies the device structure of the demultiplexers in WDM receivers but also provides a promising solution for designing other polarization-independent devices on the SiPh platform.

## Methods

**Device fabrication**. The proposed demultiplexer was fabricated on a silicon nitride platform with a $Si_3N_4$ core layer thickness of 310 nm and a buried oxide layer thickness of 2 μm. The $Si_3N_4$ layer was deposited with $SiH_4$ and $N_2$ through plasma-enhanced chemical vapor deposition. Waveguides with a width of 800 nm and EDG patterns were defined using electron-beam lithography followed by full etching of the $Si_3N_4$ layer through inductively coupled plasma (ICP) dry etching. A 1-μm-thick $SiO_2$ layer was deposited on the device as the top cladding layer. The SEM images for the device are shown in Fig. 3.

**Device characterization**. A wavelength-tunable laser (Santec, TSL-510), optical power meter (Yokogawa, AQ2211), polarization synthesizer (General Photonics, PSY-201), and end-face fiber coupling test systems were used for the measurements. The experimental setup is shown in Fig. 4a. In the measurements, the wavelength-tunable laser outputs light of the target wavelength to the polarization synthesizer, which controls the polarization of the light and outputs light with the target SOP to the end-face fiber coupling system, which couples the light with the EDG chip. The signals at the output end of the chip are coupled with the receiver fiber and transmitted to the optical power meter. All instrument control and data recording were realized through a computer. Reference waveguides were added to the chip for system loss measurement to normalize the EDG device losses. The reference waveguide consisted of only straight and bent waveguides, where the bending radius and length of the bent waveguide were the same as those for the EDG. With the reference waveguides, the fiber–waveguide coupling loss, straight waveguide loss, and bent waveguide loss can be filtered to obtain the normalized on-chip insertion loss of the EDG device.

## Data availability

The data that support the findings of this study are available through figshare: https://doi.org/10.6084/m9.figshare.21755537.v1.

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

## Acknowledgements

We acknowledge the funding support from the National Ministry of Science and Technology Key Research and Development Program (2018YFB2201200); National Natural Science Foundation of China (61635011); China Postdoctoral Science Foundation (2021M702869, 2022T150563). We acknowledge Zhejiang University Micro and Nano Processing Platform for providing the facility support. We thank Dr. Wei Ma, Dr. Tong Ye, Mr. Lin Han, and Mrs. Ying Huang for their fruitful discussions on the experiments conducted in this study and revisions made in the manuscript.

## Author contributions

C.L. conceived the idea, developed the device design, and performed the simulation verification, device fabrication, experimental characterization, and data analysis. C.L., B.X., and T.C. contributed to the writing of the manuscript. T.C. supervised the project.

## Competing interests

The authors declare no competing interests.
