## [Peer Review File · Communications Engineering]

Reviewers' comments:

Reviewer #1 (Remarks to the Author):

Polarization-independent wavelength demultiplexer based on single etched diffraction grating device
C. Li, B. Xiong and T. Chu

The authors present the design and experimental examination of a wavelength demultiplexer that simultaneously performs polarization demultiplexing with a $320\mu\text{m}$ by $230\mu\text{m}$ footprint. The device polarization demux relies on the different effective index of refraction for TE and TM modes within the freely propagating region of an otherwise conventional Roland grating structure. The device was fabricated via e-beam lithography on a Silicon Nitride on Si layer structure. The experimental results demonstrate good performance across 4 coarse wavelength division multiplexing wavelengths at O-band with ~ 0.5 to 2.4dB insertion loss, crosstalk $< 30\text{dB}$ and PDL of $< 1.8\text{dB}$. The measured device exhibited a wavelength shift compared to design simulations attributed to variation in SiN layer thickness.

The device is novel and of wide interest to others in the photonics/telecom community. The experimental methods are well conceived, and the results are reasonably complete. The supplemental materials are useful and provide the needed detail. Overall the manuscript and supplemental materials describe a useful innovative device with good experimental results.

Although the manuscript and supporting documents describe a novel device the authors can increase its impact by addressing the following:

- The strong sensitivity to layer thickness represents a limitation when considering deployment opportunities since all fabrication methods experience a thickness variation. Mitigation efforts should be discussed in more detail and the tradeoff of wavelength sensitivity vs polarization separation to layer thickness explored. It is noted that the supplementary materials do describe some features related to layer thickness but not sensitivity to variation which may suggest a thicker layer.
- The manuscript is unnecessarily repetitive
- The manuscript is excessively harsh on past solutions to the pol mux challenges.
- The primary document (not supplementary material) should include more details related to the design methods to allow the reader to appreciate the tradeoffs without always resorting to the supplementary materials. For example, equation 1 is poorly described the main manuscript and provides no value without additional detail in the main manuscript. Figure 1b is similarly in need of more description in the main manuscript.
- Additional explanation of the direct cause of the wavelength cross talk observed in both the simulation and experimental data. Is this just an FSR issue?
- The use of Bragg gratings behind the grating teeth should be referenced.

Reviewer #2 (Remarks to the Author):

This manuscript presents a novel polarization-independent wavelength demultiplexer based on an etched diffraction grating on silicon photonic platform. By utilizing birefringence effect, arbitrarily polarized light containing four wavelengths was split into eight channels with single polarization (TE/TM) and single wavelength. The design principle, simulation, test results were explained in detail. The results are novel and convincing. I believe this paper is suitable to be published in Communications Engineering.

Reviewer #3 (Remarks to the Author):

In this paper, a polarization-independent wavelength demultiplexer based on a single etched diffraction grating device is reported. An etched diffraction grating was used to split optical modes in 8 different channels which is very interesting. The results reported are quite promising and worth publishing with minor revisions.

The following parts need to improve:

1. Does the Bragg grating restrain the bandwidth of the device? The author needs to elaborate on the definition of Bragg grating and etch grating in this paper.
2. How to maintain an equal intensity of the TE and TM in each channel, since the optical intensity of the diffraction is varying with the order of the diffraction grating?
3. I notice that the input WG is with a fixed angle, what is the design consideration of the input WG angle and the position of the output WG?
4. What is the splitting ratio for the TE and TM modes?
5. In Fig. 3, the tapered WGs at the output port have different taper angles, is there any design purpose for these features?

Detailed response to the Reviewers' comments

Reviewer #1:

The authors present the design and experimental examination of a wavelength demultiplexer that simultaneously performs polarization demultiplexing with a 320 um by 230 um footprint. The device polarization demux relies on the different effective index of refraction for TE and TM modes within the freely propagating region of an otherwise conventional Roland grating structure. The device was fabricated via e-beam lithography on a Silicon Nitride on Si layer structure. The experimental results demonstrate good performance across 4 coarse wavelength division multiplexing wavelengths at O-band with ~0.5 to 2.4dB insertion loss, crosstalk <30dB and PDL of <1.8dB. The measured device exhibited a wavelength shift compared to design simulations attributed to variation in SiN layer thickness. The device is novel and of wide interest to others in the photonics/telecom community. The experimental methods are well conceived, and the results are reasonably complete. The supplemental materials are useful and provide the needed detail. Overall the manuscript and supplemental materials describe a useful innovative device with good experimental results. Although the manuscript and supporting documents describe a novel device the authors can increase its impact by addressing the following:

1. The strong sensitivity to layer thickness represents a limitation when considering deployment opportunities since all fabrication methods experience a thickness variation. Mitigation efforts should be discussed in more detail and the tradeoff of wavelength sensitivity vs polarization separation to layer thickness explored. It is noted that the supplementary materials do describe some features related to layer thickness but not sensitivity to variation which may suggest a thicker layer.

Answer:

Thank you for your kind suggestions. In the original manuscript, a detailed discussion regarding certain important aspects was missing. Accordingly, we added a relevant discussion regarding the trade-off between wavelength sensitivity, efficiency, and polarization separation in **Line 186-189 on Page 11 of the Revised manuscript**. To discuss the change in wavelength sensitivity when the Si₃N₄ layer is thicker, we added simulations in the supplemental material, as shown in **Supplementary Figure 8 and Line 143 on Page 8 to Line 152 on Page 9 of the Revised Supplementary information**. According to **Supplementary Figure 8**, the wavelength sensitivity of the EDG decreases when the Si₃N₄ layer becomes thicker, but the corresponding decrease in the polarization separation bandwidth is more pronounced. When the thickness of Si₃N₄ layer increases to 325 nm, it is just enough to meet the bandwidth requirement of CWDM4. As the Si₃N₄ layer continues to increase, the polarization separation bandwidth is insufficient. Therefore, the Si₃N₄ layer thickness selection is mainly based considering the polarization separation bandwidth.

In addition, in the EDG fabrication process, the uniformity of Si₃N₄ can be controlled by optimizing the growth process, and the thickness uniformity of the LPCVD-grown Si₃N₄ layer can be usually kept within 5% (~15 nm for a 300-nm-thick Si₃N₄ layer), so the deviation range can be represented in the figure.

Supplementary Figure 8. Simulated results of the wavelength shift and polarization dispersion characteristics at different Si₃N₄ layer thicknesses.

2. The manuscript is unnecessarily repetitive.

Answer:

Thank you for pointing out this issue. We have carefully checked the manuscript and revised the overlapping parts. Such as **Line 39–40 on Page 2 and Line 43–44 on Page 2 of the Revised manuscript.**

3. The manuscript is excessively harsh on past solutions to the pol mux challenges.

Answer:

Thank you for pointing this out. We agree with your assessment that our presentation of previous work was too harsh. To address this point, we have carefully revised the Introduction, especially the introduction to previous work in **Line 36 on Page 2, Line 39 on Page 2, Line 43 on Page 2 to Line 46 on Page 3 and Line 48–51 on Page 3 of the Revised manuscript.**

4. The primary document (not supplementary material) should include more details related to the design methods to allow the reader to appreciate the tradeoffs without always resorting to the supplementary materials. For example, equation 1 is poorly described the main manuscript and provides no value without additional detail in the main manuscript. Figure 1b is similarly in need of more description in the main manuscript.

Answer:

Thank you for this suggestion. We agree with your assessment that some of the definitions in the original manuscript were too concise. Accordingly, in the revised manuscript we have introduced into more details regarding these definitions and added trade-off considerations of the design.

We added a description for Figure 1b in **Line 100 on Page 5 to Line 105 on Page 6** of the **Revised manuscript** as follows “The grating teeth located at the tangential points of the Rowland circle (green dotted line) and grating circle are considered as the central grating teeth (0th). L_f is the Rowland circle diameter, L_{0in} and L_{0out} are the effective incident and output optical paths of the central grating tooth, respectively, L_{kin} and L_{kout} are the effective incident and output optical paths of the k th grating tooth, respectively, and θ_{in} and θ_{out} are the incidence and diffraction angles, respectively. θ_k is the rounding angle of the grating circle corresponding to the midpoint of each grating tooth.”

We also added an explanation for Eq. (1), along with Eq. (2), in **Line 105 on Page 5 to Line 113 on Page 6** of the **Revised manuscript** as follows “The EDG needs to satisfy the optical path difference, as shown in Eq. (1), and thus, θ_{out} satisfies the relationship with the effective refractive index n_{eff} of the FPR slab waveguide, as shown in Eq. (2), which is polarization-dependent. ... where m is the diffraction order, λ is the incident light wavelength, and n_{eff} is the effective index of the propagation medium, which is usually considered as the effective refractive index of the fundamental mode of the slab waveguide.”

5. Additional explanation of the direct cause of the wavelength crosstalk observed in both the simulation and experimental data. Is this just an FSR issue?

Answer:

Following your suggestion, we added the causes of wavelength crosstalk in the **Revised manuscript**. When the FSR of EDG is smaller than the operating bandwidth, the output spectrums of adjacent diffraction orders will overlap with the spectrum of the designed diffraction order, causing serious crosstalk problems, whereas our design adopts a larger FSR design (FSR = 536 nm), thus avoiding crosstalk. We respectfully believe that the sources of crosstalk in EDG are usually interference of adjacent channels coupling. As shown in the optical field energy distribution diagram (Supplementary Figures 4b and 5b), the diffraction energy of neighboring orders is very weak when $m = 3$, and the crosstalk of EDG mainly comes from the neighboring channel coupling, i.e., the energy of adjacent channels is partially coupled to the central channel, causing the crosstalk problem of the central channel. We added a description of the crosstalk source in **Lines 149-151 on Page 8** of the **Revised manuscript** as follows “The device demonstrated a good loss uniformity (approximately 0.5 dB) and low crosstalk (better than -28 dB), which is mainly caused by the coupling of adjacent channels.”

In addition, affected by polarization separation, if the thickness is not well controlled, some of the energy of TE and TM polarization will partially overlap, causing more serious wavelength

crosstalk; accordingly, we discuss this problem and adopt an appropriate thickness to avoid it and prove the feasibility of CWDM4 in a large wavelength range.

6. The use of Bragg gratings behind the grating teeth should be referenced.

Answer:

Thanks for this suggestion. Indeed, the original manuscript did not contain references regarding Bragg gratings. We have added references to the Bragg gratings behind the grating teeth in **Line 85 on Page 4** of the **Revised manuscript**.

Reviewer #2:

This manuscript presents a novel polarization-independent wavelength demultiplexer based on an etched diffraction grating on silicon photonic platform. By utilizing birefringence effect, arbitrarily polarized light containing four wavelengths was split into eight channels with single polarization (TE/TM) and single wavelength. The design principle, simulation, test results were explained in detail. The results are novel and convincing. I believe this paper is suitable to be published in *Communications Engineering*.

Thank you for your positive assessment of our manuscript and for recommending its publication in *Communications Engineering*.

Reviewer #3:

In this paper, a polarization-independent wavelength demultiplexer based on a single etched diffraction grating device is reported. An etched diffraction grating was used to split optical modes in 8 different channels which is very interesting. The results reported are quite promising and worth publishing with minor revisions.

The following parts need to improve:

To Reviewer 3:

1. Does the Bragg grating restrain the bandwidth of the device? The author needs to elaborate on the definition of Bragg grating and etch grating in this paper.

Answer:

Thank you for your kind suggestion and this very important question. We originally introduced Bragg gratings to increase the diffraction efficiency of the etched grating teeth. However, the effect on bandwidth was not considered. To explore this, we simulated the output spectrum of EDG without Bragg gratings under two polarizations and found that the overall polarization bandwidth remains basically unchanged, as shown in **Figure R1** below.

Figure R1. Simulated transmission spectrum of polarization-independent demultiplexing EDG. Output from channels 1–4 with TE input (a) without and (c) with Bragg gratings. Output from channels 5–8 with TM input (b) without and (d) with Bragg gratings.

Therefore, the limitation of Bragg gratings on the EDG bandwidth is mainly in the insertion loss performance. As the Bragg grating is used to increase the reflectivity of the etched grating, it has an inherent working bandwidth, which is required to be larger than the working bandwidth of the device and therefore requires additional design consideration, as shown in **Supplementary Figure 3**.

In the original manuscript, the Bragg gratings and etched gratings definitions were indeed missing, thank you for pointing this out. We added these definitions in **Line 81–86 on Page 4** of the **Revised manuscript** as follows “The proposed EDG device is fabricated on a Si_3N_4 thin-film platform consisting of four main components, namely, input waveguides for input signal light, FPR for light transmission, etched gratings for reflecting and converging multi-wavelength multi-polarization signal lights, and TE/TM output waveguides for outputting ideal signal lights. In addition, one-dimensional Bragg gratings^{32,33}, which are typically used to increase the reflection efficiency of EDG teeth, are added behind the grating teeth.”

2. How to maintain an equal intensity of the TE and TM in each channel, since the optical intensity of the diffraction is varying with the order of the diffraction grating?

Answer:

Thanks for raising such an important issue. The diffracted light intensity does vary with the number of diffraction orders. How to ensure low loss, low PDL, and better loss uniformity of the TE and TM channels are some of the main issues considered in our design.

First, we need to ensure that the Bragg gratings can work in both TE and TM polarizations simultaneously, providing a comparably high reflection efficiency of the grating teeth for both

polarization signals, thus ensuring that the signal intensity is not affected by the reflection efficiency of the grating teeth.

Subsequently, we simulated and investigated the variation in TE and TM output spectra for EDG with multiple diffraction order designs. As shown in **Supplementary Figures 4 and 5**, the increase in diffraction order leads to an increase in loss under TE and TM polarization. Therefore, we choose an appropriate structure design so that the device operates at the same diffraction order in both polarizations and achieves high efficiency.

Supplementary Figure 4. (a) Output spectra of the central output channel under TE polarization for different diffraction orders. (b) Electric field distribution at $\lambda = 1311$ nm for different diffraction orders.

Supplementary Figure 5. (a) Output spectra of the central output channel under TM polarization for different diffraction orders. (b) Electric field distributions at $\lambda = 1311$ nm for different diffraction orders.

In addition, although the same diffraction order is controlled, a certain intensity nonuniformity also exists in different channels due to the presence of aberrations. To achieve higher channel uniformity, we simulated the output spectra at different diffraction orders and observed that smaller diffraction orders can also achieve better intensity uniformity. Therefore, our design uses a smaller number of diffraction order, which results in higher uniformity (less than 0.5 dB).

Figure R2. (a) Output spectra of output channels 1–4 under TE polarization for different diffraction orders. (b) Output spectra of output channels 5–8 under TM polarization for different diffraction orders.

3. I notice that the input WG is with a fixed angle, what is the design consideration of the input WG angle and the position of the output WG?

Answer:

We set the input waveguide angle to 45° , which is commonly used in the design of etched diffraction gratings. This is because in the Fresnel Kirchhoff diffraction law, the diffracted light energy of the light in the exit direction is proportional to $\cos(\theta_{in}) + \cos(\theta_{out})$. Therefore, when the input angle is set to 45° , it is conducive to achieving higher diffraction efficiency and lower insertion loss. In addition, the 45° diffraction angle can balance the EDG tooth size and increase the process fabrication tolerance.

First, we calculate the position of the output waveguide at the central wavelength. The positions of other output waveguides are determined by the waveguide spacing. We determined the appropriate waveguide spacing through the simulation of different waveguide spacings, as shown in **Supplementary Figure 6**, so as to determine the position of the output point in the Rowland circle.

4. What is the splitting ratio for the TE and TM modes?

Answer:

For wavelength and polarization multi-dimensional multiplexed EDG, the separation ratio of the TE and TM polarization modes is similar to the extinction ratio parameter, and according to the polarization extinction ratio of PBS, we define the TE and TM extinction ratios as follows.

The TE extinction ratio at four wavelengths is defined as the difference between the output power of output waveguides 1, 2, 3, and 4 and that of output waveguides 5, 6, 7, and 8, respectively, under TE polarization.

$$ER_{TE, \lambda_1/\lambda_2/\lambda_3/\lambda_4} = 10 \log_{10} (T_{output1/2/3/4, TE} / T_{output5/6/7/8, TE})$$

The TM extinction ratio at four wavelengths is defined as the difference between the output power of output waveguides 5, 6, 7, and 8 and that of output waveguides 1, 2, 3, and 4, respectively, under TM polarization.

$$ER_{TM,\lambda_1/\lambda_2/\lambda_3/\lambda_4} = 10 \text{Log}_{10} (T_{output5/6/7/8,TM} / T_{output1/2/3/4,TM})$$

Therefore, according to our experimental results, as shown in **Figure 4**, the extinction ratio of the prepared EDG is better than -20 and -25 dB under the TE and TM polarizations.

Fig. 4. EDG measurements. **a** Experimental setup. **b** Optical microscope view of the fabricated EDG. **c–f** Transmission spectra. **c** Outputs from channels 1–4 with TE input. **d** Outputs from channels 5–8 with TE input. **e** Outputs from channels 1–4 with TM input. **f** Outputs from channels 5–8 with TM input.

5. In Fig. 3, the tapered WGs at the output port have different taper angles, is there any design purpose for these features?

Answer:

We greatly appreciate this question from reviewer. In our design, after determining the output position, we use the output position as the center point of the taper and the line between the output position and the center grating tooth surface as the direction of the output waveguide, thus determining the taper parameters to obtain the highest possible efficiency. Therefore, the difference in the output position of each wavelength causes the difference in angle.

In addition, to further reduce the influence of polarization dispersion characteristics, we appropriately reduce the width of the middle two tapers to avoid the mutual interference of TE and TM polarizations.

REVIEWERS' COMMENTS:

Reviewer #1 (Remarks to the Author):

The revised manuscript has resolved the issues raised and the manuscript is improved. The device is novel and potentially useful. There is sufficient information for the reader to understand and build on this work. I believe it is now suitable for publication.

Reviewer #2 (Remarks to the Author):

The points raised in the previous round of review have been satisfactorily addressed. I believe this paper is suitable to be published in Communications Engineering.

Reviewer #3 (Remarks to the Author):

All my comments have been addressed and I appreciate that the authors did some extra work to address these comments. Thanks